# Diaphragmatic Mobility and Chest Expansion in Patients with Scapulocostal Syndrome: A Cross-Sectional Study

**DOI:** 10.3390/healthcare10050950

**Published:** 2022-05-20

**Authors:** Thanaporn Srijessadarak, Preeda Arayawichanon, Jaturat Kanpittaya, Yodchai Boonprakob

**Affiliations:** 1Faculty of Associated Medical Sciences, School of Physical Therapy, Khon Kaen University, Khon Kaen 40002, Thailand; sr_thanaporn@kkumail.com; 2Faculty of Medicine, Research Institute for Human High Performance and Health Promotion, Khon Kaen University, Khon Kaen 40002, Thailand; 3Department of Rehabilitation Medicine, Faculty of Medicine, Khon Kaen University, Khon Kaen 40002, Thailand; prearr@yahoo.com; 4Department of Radiology, Faculty of Medicine, Srinagarind Hospital, Khon Kaen University, Khon Kaen 40002, Thailand; jatkan@gmail.com; 5Department of Physical Therapy, Faculty of Associated Medical Science, Khon Kaen University, Khon Kaen 40002, Thailand

**Keywords:** scapulocostal syndrome, myofascial pain syndrome, respiratory characteristics, diaphragmatic mobility, chest expansion

## Abstract

Scapulocostal syndrome (SCS) is a subset of myofascial pain syndrome affecting the posterior shoulder and upper back area. Some of the affected muscles are attached to the rib cage, which may affect diaphragmatic mobility and chest expansion. The purpose of this study was to investigate the characteristics of diaphragmatic mobility and chest expansion in patients with SCS. Twenty-nine patients with SCS and twenty-nine healthy participants of a similar age, gender, weight, and height were included in the study. All participants were evaluated for diaphragmatic mobility (DM) by real-time ultrasound (RTUS) and for chest expansion (CE) using a cloth tape measure. An independent t-test was used to compare the outcome variables between groups. The DM value in the SCS group was 46.24 ± 7.26 mm, whereas in the healthy group it was 54.18 ± 9.74 mm. The DM value was lower in the SCS group compared to in healthy participants (*p* < 0.05). Chest expansion at the axilla, the fourth intercostal space (4th ICS), and the xiphoid level in the SCS group was 7.26 ± 1.13, 6.83 ± 0.94, and 6.86 ± 1.25, respectively, while chest expansion at the axilla, 4th ICS, and xiphoid level in the healthy group was 7.92 ± 1.39, 7.54 ± 1.43, and 8.13 ± 1.32, respectively. Chest expansion at the 4th ICS and the xiphoid level in the SCS group was significantly lower than in the healthy group (*p* < 0.05). Patients with SCS presented a decrease in diaphragmatic mobility and chest expansion. Therefore, SCS treatment programs ought to add breathing exercises to improve lung expansion.

## 1. Introduction

Scapulocostal syndrome (SCS) is a chronic myofascial pain syndrome affecting the thoracic and scapular regions. SCS pain is ongoing and usually lasts longer than three months [1]. The thoracic spine pain prevalence data obtained from 1 year ranged from 3.0 to 55.0% [2]. The prevalence of lifetime upper back pain was found to be 59.5% [3]. Moreover, the highest incidence of SCS was found in middle-aged people between 18 and 60 years old and was especially prominent in the adult working population. This syndrome is found in females more than in males. Poor sitting posture while working or using digital media are risk factors for SCS, as it is an overuse disorder caused by the repeated inadequate use of the muscles around the scapulae. This syndrome can be found at myofascial trigger points (MTrPs) on the muscles around the scapulae, including the levator scapulae, upper trapezius, rhomboid major and minor, teres major and minor, infraspinatus, serratus anterior, and serratus posterior superior muscles. Hence, this syndrome affects the biomechanics of the scapulae [1].

Some of the affected muscles are attached to the rib cage, which may affect chest expansion during breathing. To elucidate, the close anatomical, musculoskeletal, and neural associations of the scapular region with the thoracic spine may cause biomechanical alterations in the thoracic spine and rib cage. Postural muscles have two main functions: postural control and respiration. Interestingly, the scapular muscles are a subset of the postural muscles, and they provide the control of upper back posture and upper chest breathing [4]. The scapular muscles may directly affect respiration. In addition, the scapular muscles connect to the inner and outer core stabilizers, which indirectly affect chest expansion and diaphragmatic mobility [5,6]. The core comprises various muscles that stabilize the shoulders, the pelvis, and the spine and provides a base for the movement of the limbs. The major core muscles include the transversus abdominus in the anterior, the multifidus in the posterior, the pelvic floor in the inferior, and the diaphragm in the superior. The minor core muscles are the latissimus dorsi, the gluteus maximus, and the trapezius. All these muscles connect directly or indirectly to the thoracolumbar fascia and the spinal column, and they are also responsible for attaching the upper and lower extremities. The core is considered the center of the functional kinetic chain. The core muscles are activated through a feed-forward mechanism shortly before movements of the upper and lower limbs, and they act as a base to support the performance of skilled movements. This feed-forward mechanism is essential for attaining mobility and the stability of the extremities. These findings encourage the theory that movement control and stability are developed in a core-to-extremity (proximal-distal) and cephalocaudal (head-to-toe) manner [7]. Therefore, patients with SCS should be investigated with regard to their respiratory characteristics.

Recently, respiration and the function of the diaphragm muscle have been evaluated in many myofascial pain syndromes, including neck pain, temporomandibular joint pain, low back pain, and lumbopelvic pain [8]. Janssens and coworkers found that participants with low back pain presented with more diaphragm fatigability when compared with healthy participants [9]. Mohan et al. showed that diaphragmatic movement and respiratory muscle endurance were poorer in the nonspecific lower back pain group than in the healthy group [10]. Recently, Calvo-Lobo and colleague reported that participants with lumbopelvic pain had a reduced diaphragm thickness compared to healthy matched-paired participants [11]. The diaphragm muscle is one of the core stabilizers related to postural control. An ineffective diaphragm muscle leads to poor postural control, poor balance, adjusted proprioception, and ineffective motor control. Additionally, this leads to abnormal breathing, which is characterized by using the accessory muscles of respiration, including the sternocleidomastoid, upper trapezius, and scalene muscles. The over-action of these accessory muscles causes neck pain, scapular dyskinesis, and trigger point formation [8]. Moreover, Ahmad and colleagues in 2022 found that FHP was correlated with a decrease in respiratory muscle strength in patients with chronic neck pain. This was caused by morphological and biomechanical changes in the thoracic cage. To clarify, FHP led to the expansion of the upper chest and the narrowing of the lower chest, which limited lower chest expansion. In addition, FHP contributed to abdominal muscle shortening, resulting in a decrease in the anteroposterior diameter of the lower chest. This limited diaphragmatic mobility [12].

The function of the diaphragm can be measured during inhalation and exhalation. There are various procedures to evaluate diaphragmatic mobility, including fluoroscopy, respiratory muscle strength, X-ray, spirometry, lower chest expansion, and real-time ultrasound [13]. Fluoroscopy is the standard method used for assessing the position and mobility of the diaphragm. However, this method has a drawback: patients evaluated through fluoroscopy are exposed to ionizing radiation [14]. Evaluations of lower chest expansion, respiratory muscle strength, and spirometry are indirect procedures used to reveal diaphragmatic mobility and expose the function of other respiratory muscles, such as the abdominal, intercostal, and accessory respiratory muscles [15].

Chest expansion is used to assess rib cage mobility and is associated with lung volume. Upper and lower CE are commonly used in clinical practice to assess chest expansion and to present indirect data on lung function [16]. An association between upper or lower CE and maximal inspiratory pressure has been determined in patients with fibromyalgia and osteoporosis. Anatomical references for upper CE include the fourth intercostal space, the axillary level, and the 5th thoracic vertebrae, and references for lower CE include the xiphoid level and the 10th thoracic vertebrae. As CE is measured using a cloth tape measure, it is a simple, inexpensive, and noninvasive method for evaluating chest mobility [17]. Previous studies have found that lower chest expansion is correlated with diaphragmatic mobility (r = 0.74, *p*-value = 0.001) [18]. Consequently, a cloth tape measure was deemed to be a suitable instrument for evaluating chest expansion in this study.

Real-time ultrasound is commonly employed to assess the role of various essential internal organs, including the heart, colon, kidneys, spleen, and liver [19]. The advantage of this method is that it is non-invasive and free of ionizing radiation. Additionally, the information from real-time ultrasound can be stored for subsequent consideration [20,21]. Consequently, real-time ultrasound was deemed to be a suitable instrument for evaluating the mobility of the diaphragm in this study.

The purpose of this study was to investigate the characteristics of diaphragmatic mobility and chest expansion in patients with SCS. These data could be important for future studies. If SCS and breathing are related, it is relevant to treat SCS to improve respiratory function. On the other hand, patients with SCS ought to be treated with diaphragmatic training.

## 2. Materials and Methods

### 2.1. Study Design

This cross-sectional study recruited participants from Khon Kaen University, Thailand. The study was approved by the Khon Kaen University Ethics Committee for Human Research (HE631436). The sample size was calculated by the outcome (diaphragmatic mobility) from the results of the study conducted by Mohan and coworkers [10]. They reported diaphragmatic mobility for 34 participants with nonspecific low back pain and 34 healthy controls. The diaphragmatic mobility in participants with nonspecific low back pain was 45.09 mm with a standard deviation of 9.89, and the diaphragmatic mobility in healthy controls was 50.09 mm with a standard deviation of 9.18. A pooled variance estimate (σ^2^) for calculating the sample size was used as follows:$$\sigma^{2}=\frac{\left(n_{1}-1\right){\left(s_{1}\right)}^{2}+\left(n_{2}-1\right){\left(s_{2}\right)}^{2}}{n_{1}+n_{2}-2}$$

The pooled variance estimate (σ^2^) was 91.04. In 2017, Mohan and colleagues reported that the clinical significance of diaphragmatic mobility was 7.09 [22]. A significance level of lower than 0.05 (Z_α/2(0.025)_ = 1.96) and a power of 80% (Z_β(0.1)_ = 0.84) were used to calculate the sample size as follows:$$n/\mathrm{group}=\frac{2{\left(Z_{\alpha }+Z_{\beta }\right)}^{2}{\left(\sigma \right)}^{2}}{{\left(\mu_{1}-\mu_{2}\right)}^{2}}$$

A total of 29 patients with SCS as well as another 29 healthy participants were included.

### 2.2. Participants

The participants with SCS, aged 18–50 years with normal BMI (18.5–22.9 kg/m^2^), were recruited via bulletin boards in Khon Kaen province in addition to verbal requests for participants during a 6-month period between December 2020 and May 2021. Health history and a physical examination were recorded and performed respectively by a physiatrist at Srinagarind Hospital in order to confirm each patient’s diagnosis. Participants were included if they had experienced pain at the scapular region for longer than 12 weeks (VAS equal to or more than 5 cm) covering at least one MTrP in the muscles around the scapular region, i.e., the levator scapulae, trapezius, rhomboid, teres, and serratus posterior superior muscles. The MTrP diagnostic criteria was based on research by Simons and coworkers [23]. Trigger points were detected by the presence of a tender spot within the palpable taut bands of muscle in the regions that the patient identified as painful. MTrPs produce a pain referral pattern [24]. Healthy participants were recruited to the control group if they had no history of SCS throughout the 12 months prior.

The exclusion criteria incorporated any of the following disorders: history of degenerative shoulder joint disease, rotator cuff disease, adhesive shoulder capsulitis, cervical radiculopathy with facet joint dysfunction and/or intervertebral disc herniation, lumbar intervertebral disc herniation, lumbar stenosis, lumbar spondylosis, lumbar spondylolisthesis, history of radiotherapy, chronic respiratory diseases (chronic obstructive pulmonary disease (COPD), asthma, occupational lung diseases, or pulmonary hypertension), smoker, and ex-smoker.

### 2.3. Procedures

#### 2.3.1. Diaphragmatic Mobility

Diaphragmatic mobility was evaluated via real-time ultrasound (RTUS) (Sonoscape ultrasound) with a 2–4 MHz convex transducer. This tool is valuable for accurately assessing diaphragmatic mobility. RTUS showed a high current validity (r = 0.78 to r = 0.83) [25]. Each participant laid on their back with their head elevated at 30 degrees. The assessor placed the transducer over the right subcostal area, with the striking angle of the ultrasound towards the cranio-caudal axis, to identify the left portal vein branch as a reference point. This is considered a valid reference point [26]. The assessor recorded the distance between the highest point of the right hemidiaphragm during maximal inhalation and at the end of maximal exhalation (Figure 1), with values taken in millimeters [10]. The measurement was carried out three times with the maximum value applied.

#### 2.3.2. Chest Expansion

Chest expansion was evaluated in the sitting position using a cloth tape measure. It was evaluated in 3 positions: the axilla, the fourth intercostal, and the xiphoid level. The assessor recorded an average of the distance between the maximal inspiratory maneuver and at the end of the maximal expiratory maneuver. The measurement was carried out twice with values taken in centimeters [10].

### 2.4. Statistical Analysis

All statistical analyses were performed using the SPSS version 26 statistical software. Results were presented as a mean ± standard deviation (SD). DM and chest expansion demonstrated a normal distribution based on the Shapiro–Wilk test with *p* > 0.05. Independent *t*-tests were used to compare all variables. For all tests, the statistical significance threshold was set at *p* < 0.05.

## 3. Results

### 3.1. Demographic Data and Baseline Clinical Characteristics

The demographic data and baseline clinical characteristics of SCS participants and healthy participants are shown in Table 1. Twenty-nine SCS participants (17 female and 12 male) were matched with twenty-nine healthy participants (17 female and 12 male). There was no significant difference between demographic data, including age, height, weight, and BMI, with *p* > 0.05. In the SCS group, the MTrPs were found in the levator scapulae, upper trapezius, and rhomboid muscles. The number of participants who exercised in the SCS group was lower than that in the healthy group.

### 3.2. Diaphragmatic Mobility and Chest Expansion between the SCS Group and the Healthy Group

Demographic data of respiratory characteristics in this study are presented in Table 2. The DM value in the SCS group was 46.24 ± 7.26 mm, whereas in the healthy group it was 54.18 ± 9.74 mm. The difference in value between groups was −7.94 (95% CI −12.46 to −3.41). The DM value was lower in the SCS group when compared to healthy participants (*p* < 0.05). Chest expansion at the axilla, 4th ICS, and xiphoid level in the SCS group was 7.26 ± 1.13, 6.83 ± 0.94, and 6.86 ± 1.25, respectively, while chest expansion at the axilla, 4th ICS, and xiphoid level in the healthy group was 7.92 ± 1.39, 7.54 ± 1.43, and 8.13 ± 1.32, respectively. Chest expansion at the 4th ICS and xiphoid level in the SCS group was significantly lower than in the healthy group (*p* < 0.05).

## 4. Discussion

This study evaluated diaphragmatic mobility and chest expansion in patients with SCS compared to healthy matched-paired participants. The results of this study demonstrated that SCS patients showed less diaphragmatic mobility and chest expansion at the 4th ICS and xiphoid level than healthy participants. Nevertheless, there were no differences in chest expansion at the axilla level.

Reduced diaphragmatic mobility and chest expansion in SCS patients were found in this study. To our knowledge, this is the first study to assess diaphragmatic mobility and chest expansion in patients with SCS. However, respiratory characteristics have been evaluated in patients with neck pain, low back pain, and lumbopelvic pain [8]. The possible mechanism for explaining these results will be discussed in a logical way as follows:

Normal breathing includes coordinated movement of the upper chest, lower chest, and abdomen. Furthermore, normal breathing requires the sufficient use and functionality of the diaphragm muscle. Anatomically, the diaphragm muscle is located in the lower chest. During inhalation, the abdomen moves forward as the lower six ribs laterally expand, elevate, and rotate upward relative to the spine. The sternum and the remainder of the thoracic cavity move anteriorly and superiorly, expanding the chest volume as the diaphragm descends. This produces a negative pressure gradient to draw air into the lungs. During expiration, the diaphragm relaxes and returns to a dome shape [8]. Therefore, reduced diaphragmatic mobility in this study limited expansion of the middle and lower chest, while the upper chest was not affected.

This study revealed that chest expansion in SCS patients is inferior compared to that in healthy participants. To clarify, SCS is a chronic myofascial pain syndrome, and the myofascial trigger points (MTrPs) can be found at the muscles affected by SCS, including the levator scapulae, upper trapezius, serratus posterior superior, serratus anterior, rhomboids major and minor, infraspinatus, and teres minor [27]. Some theories have considered the association between MTrPs and joint hypomobility. To explicate, the increased muscle tension caused by MTrPs can lead to displacement stress on the joint. Hence, MTrPs provoke joint dysfunction [28]. In this way, MTrPs lead to muscle weakness and muscle tightness, thus contributing to the decreased stability of the cervical and thoracic spine as well as causing changes to rib cage mechanics [29]. The spasm of the scapular muscles causes the elevation of the upper chest. This leads to the diaphragm being in a low position. Then, exhalation is limited due to the elevation of the upper chest [30]. This also limits the expansion of the middle and lower chest. This situation causes the reduced mobility of the diaphragm.

This explanation has been confirmed by investigating diaphragmatic mobility using real-time ultrasound, which directly measures diaphragmatic mobility [18]. The results of this study demonstrate that diaphragmatic mobility in patients with SCS is lower than in healthy participants. Some of the affected SCS muscles attach at the rib cage and connect to the diaphragm muscle through core muscles [5]. Imagine that the abdominal cavity is a house: the roof is the diaphragm and the floor is the pelvic floor muscle. The wall that surrounds the house is the transversus abdominis (TA), and the rivets of the walls are the multifidus muscles [30]. The diaphragm and pelvic floor muscles work in synergy with the TA, and they are responsible for sustaining and increasing intra-abdominal pressure during various postural tasks [31]. Hodges and Gandevia discovered that, during repetitive arm flexions in the standing position, the contraction of the TA occurred prior to the initiation of an arm movement and substantial contraction of the diaphragm and pelvic floor muscles. On the other hand, some studies have reported that the activation of the diaphragm transpires prior to upper limb movement and occurs simultaneously with the activation of the TA [32]. Based on myofascial linkages, these muscles are linked muscle chains that produce trunk stability during movement and force transmission from the lower to upper limbs [33]. The thoracolumbar fascia is connected to the internal and external oblique, transverse abdominis, latissimus dorsi, and gluteal maximus, which distribute the load between the upper and lower extremities. Furthermore, the latissimus dorsi muscle is connected at the inferior border of the scapulae. SCS leads to scapular hypomobility, which may affect the latissimus dorsi [5,34]. Therefore, pain at the affected SCS muscles may lead to difficulty amid the movement of the diaphragm muscle through the core muscles.

This is confirmed by a previous study, whereby the use of electronic devices in a poor posture (forward head posture, round shoulders, and kyphosis) was also found to be a risk factor for scapulocostal syndrome. The characteristics include a prolonged sitting posture, an awkward posture, and repetitive movement. An awkward posture is defined by the simultaneous combination of forward head posture and round shoulders. The cervical and thoracic regions of the spine are held in flexion, which stimulates the over-contraction of the dorsal muscles, such as the neck extensor and upper back extensor muscles. Thus, an awkward posture may contribute to neck and shoulder pain. These factors may be caused by a muscle imbalance in the upper back or upper crossed syndrome. This is associated with tightness in the pectoral and neck extensor muscles. Weakness in the neck flexor and interscapular muscles may occur simultaneously. This weakness is usually caused by guarding, without atrophy or neurophysiologic evidence of denervation on electromyography [1], and it usually involves scapular stabilizers consisting of trapezius and serratus anterior muscles. In addition, weakness in the levator scapulae or rhomboid muscles presents along with MTrPs [35]. These postures increase intra-abdominal pressure, making it difficult to move the diaphragm [36]. Similarly, in participants with forward head posture and torticollis, decreasing the cervical curve and round shoulders compresses the chest cavity, which can alter respiratory capacity [37,38]. A recent study revealed that participants with poor posture, neck pain, lower back pain, scapular dyskinesis, and temporomandibular joint pain presented with signs of abnormal breathing mechanics [8]. Changes in cervicothoracic mobility influences an abnormal respiration pattern by decreasing the movement and strength of the diaphragm [39]. Moreover, these current findings were consistent with a study by Mohan and colleagues, who investigated respiratory characteristics in individuals with non-specific low back pain (NS-LBP). They found that there were signs of abnormal breathing in the NS-LBP patients compared to the healthy participants. Diaphragmatic mobility and respiratory muscle endurance were found to be inferior in the NS-LBP group. Chest expansion revealed a significant decrease at the level of the fourth intercostal space in the NS-LBP group [10].

This study presents certain limitations. Left hemi-diaphragmatic mobility was not assessed in this study. Thus, future studies ought to evaluate both sides of diaphragmatic mobility in patients with SCS, as the zone of apposition (ZOA) at both sides is not the same. In future studies, respiratory characteristics should be measured by a spirometer as a gold standard measure of the respiratory function. Moreover, future studies should evaluate the number of trigger points for each participant. There may be a strong correlation between the number of trigger points and the amount of diaphragmatic excursion.

## 5. Conclusions

SCS patients presented diaphragmatic mobility and chest expansion inferior to that of healthy participants. This study suggests that patients with SCS should be investigated with regard to their diaphragmatic mobility and chest expansion. Moreover, this study also recommends that treatment concerning SCS should focus on diaphragmatic training in future studies.

## Figures and Tables

**Figure 1 healthcare-10-00950-f001:**
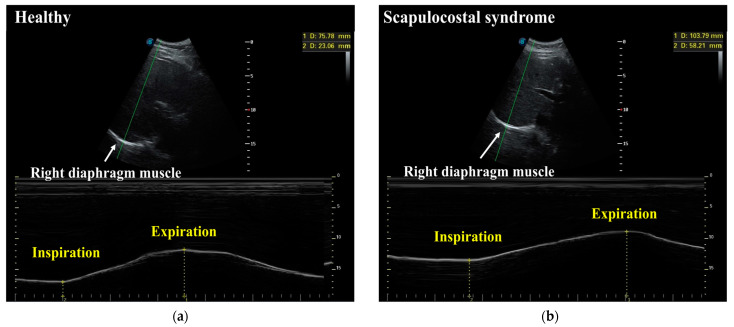
Example of comparison of diaphragmatic mobility using real-time ultrasound in (**a**) a healthy participant, and (**b**) a scapulocostal syndrome patient.

**Table 1 healthcare-10-00950-t001:** Demographic data of the sample population at baseline.

Characteristics	SCS Group (*n* = 29)	Healthy Group (*n* = 29)	*p* Value
	Mean ± SD	Mean ± SD	
Age (years)	26.86 ± 4.22	26.86 ± 4.22	1.000
Gender (female:male)	17:12	17:12	
Weight (kg)	56.36 ± 8.57	56.17 ± 8.48	0.933
Height (cm)	164.00 ± 9.87	164.55 ± 9.32	0.828
BMI (kg/m^2^)	20.86 ± 1.46	20.64 ± 1.40	0.566
Affected muscle			
Levator scapulae (%)	10 (34.48%)	0	
Upper trapezius (%)	8 (27.59%)	0	
Rhomboid (%)	11 (37.93)	0	
Exercise (yes:no)	13:16	23:6	

SCS: scapulocostal syndrome; SD: standard deviation; kg: kilogram; cm: centimeter; m: meter.

**Table 2 healthcare-10-00950-t002:** Diaphragmatic mobility and chest expansion between the SCS group and the healthy group.

Characteristics	SCS Group (*n* = 29)	Healthy Group (*n* = 29)	Difference (95% CI)	*p* Value
	Mean ± SD	Mean ± SD		
Diaphragmatic mobility (mm)	46.24 ± 7.26	54.18 ± 9.74	−7.94 (−12.46 to −3.41)	0.001 *
Chest expansion (cm)				
1. Axilla: mean	7.26 ± 1.13	7.92 ± 1.39	−0.66 (−1.33 to 0.01)	0.053
2. 4th ICS: mean	6.83 ± 0.94	7.54 ± 1.43	−0.71 (−1.34 to −0.07)	0.031 *
3. Xiphoid: mean	6.86 ± 1.25	8.13 ± 1.32	−1.27 (−1.95 to −0.60)	<0.001 *

* Statistically significant (*p* < 0.05). SCS: scapulocostal syndrome; SD: standard deviation; mm: millimeter; cm: centimeter; ICS: intercostal space.

## Data Availability

Not applicable.

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
