# Peer review of "Diaphragmatic Mobility and Chest Expansion in Patients with Scapulocostal Syndrome: A Cross-Sectional Study"

_healthcare, 2022, doi:10.3390/healthcare10050950_

Round 1

Reviewer 1 Report

  1. Line 33: The definition of the scapulocostal syndrome is vague. Is it similar to “upper cross syndrome”? Poor seated posture during working frequently causes upper cross syndrome, which involves tight upper trapezius and levator scapulae with inhibited rhomboideus and serratus anterior. Why does the scapulocostal syndrome, sharing similar risk factors with upper cross syndrome, characterized to have myofascial trigger points at rhomboideus and serratus anterior?
  2. Line 82: The authors cited reference #7 to perform sample size calculation. However, reference #7 is just a comment letter, not a study. Furthermore, the authors should indicate the statistical software they used to calculate the sample size, the effect size, the alpha error probability and power if they want to calculate the sample size for this study.
  3. Line 92: Is the scapulocostal syndrome sufficiently defined by just one myofascial trigger point around the scapular region?
  4. Line 96: Why did the study not exclude those with history of radiotherapy? Patients received radiotherapy can have much reduced chest cage mobility.
  5. Line 106: Is it calculated by M mode ultrasonography? The right subcostal area is large, and different location of probe placement may greatly impact the diaphragmatic excursion. How did the authors reduce the variations of probe placement among different participants?
  6. Line 127: Was your data normally distributed calculated by the Shapiro-Wilk test? If it was not, how could you use independent t test to perform statistical analysis?
  7. Line 134: The content in the paragraph should not redundantly duplicate the data in the tables.
  8. Line 141: The authors should use dot plot to present their data grouped by the number of trigger points for each participants. Is there a high correlation between the number of trigger points and the amount of diaphragmatic excursion?

Author Response

The school of Physical Therapy,

Faculty of Associated Medical Science,

Khon Kaen University, Khon Kaen 40002,

Thailand Tel / Fax 66-43-202085 Email: [email protected]

Date 6 May 2022

Dear Ms. Alina-Sabina Buglea

Editor-in-Chief of Healthcare– MDPI

Title: Selected respiratory characteristics in patients with scapulocostal syndrome: A cross-sectional study

Authors: Thanaporn Srijessadarak, Preeda Arayawichanon, Jaturat Kanpittaya and Yodchai Boonprakob

I am pleased to submit an original research article entitled " Selected respiratory characteristics in patients with scapulocostal syndrome: A cross-sectional study " for publication in the Healthcare. We investigated diaphragmatic mobility and chest expansion between individuals with and without scapulocostal syndrome. The result showed that diaphragmatic mobility value was lower in the SCS group when compared to healthy participants. Chest expansion at the 4th ICS and xiphoid level in the SCS group was significantly less than in the healthy group. In conclusion, SCS patients presented diaphragmatic mobility and chest expansion less than healthy participants. Therefore, these variables should be investigated in patients with SCS, and SCS treatment should focus on diaphragmatic mobility and chest expansion. We believe these findings will be interest to the reader of your journal.

On behalf of all authors, I would like to sincerely thank you for all kind correspondence and support from the editor. I also thank to valuable time, comments and suggestions from the editor and reviewers helping to improve the quality and clarity of the paper. In this version, the major and minor revisions have been made according to the suggestion as indicated using yellow highlight and explained in the “response to reviewers” file. If there is any further information about my work, please do not hesitate to contact me. I am looking forward to hearing from you very soon.

Yours sincerely,

Yodchai Boonprakob

Response to Reviewer 1 Comments

Point 1: 1. Line 33: The definition of the scapulocostal syndrome is vague. Is it similar to “upper cross syndrome”? Poor seated posture during working frequently causes upper cross syndrome, which involves tight upper trapezius and levator scapulae with inhibited rhomboideus and serratus anterior. Why does the scapulocostal syndrome, sharing similar risk factors with upper cross syndrome, characterized to have myofascial trigger points at rhomboideus and serratus anterior?

Response 1: Muscle imbalance is a time-dependent disorder which is progressed gradually. According to Janda’s concept, imbalance of muscle length or strength between agonistic muscles and antagonistic muscles are the key components of muscle imbalance. Janda’s concept was proposed the possible neurological mechanism. For instance, when pectoral muscles are in the locked short or concentric contraction. These muscles send the inhibitory signaling to reduce the contraction of antagonistic muscles by reciprocal inhibition. Subsequently, muscle weakness of the scapula, shoulder, and arm may occur (Page et al., 2010). It may be involved by the biomechanics of activities daily living or working position. The characteristics are a prolonged sitting posture, an awkward posture, and a repetitive movement (Chaiklieng et al., 2010; Feng et al., 2014; Hayes et al., 2013). Muscle weakness and muscle tightness of UCS can be found MTrP. For example, inter scapular muscles overlap with the affected muscle of SCS, which is the same function. Therefore, scapulocostal syndrome had similar risk factors to the upper cross syndrome.

Point 2: 2. Line 82: The authors cited reference #7 to perform sample size calculation. However, reference #7 is just a comment letter, not a study. Furthermore, the authors should indicate the statistical software they used to calculate the sample size, the effect size, the alpha error probability and power if they want to calculate the sample size for this study.

Response 2: The sample size calculation was rewriting followed by the suggestion of reviewer as shown in the materials and methods and started from page 3 Line 128 and reference started from page 8 Line 340.

The sample size was calculated by an outcome (diaphragmatic mobility) from the results of a previous study [10]. Mohan and coworkers reported diaphragm mobility of 34 par-ticipants with nonspecific low back pain and 34 as healthy controls. The diaphragm mobility in participants with nonspecific low back pain was 45.09 mm. with standard deviation of 9.89 and the diaphragm mobility in healthy controls was 50.09 mm. with standard deviation of 9.18. A pooled variance estimate (σ2) is 91.04. Previous study report that the clinically significant of diaphragm mobility was 7.09 [10]. The significant level of lower than 0.05 (Zα/2 (0.025) = 1.96) and a power of test at 80% (Zβ (0.1) = 0.84) were used to calculate. A total of 29 patients with SCS as well as another 29 healthy participants were included.

Refrerence: Mohan,V.; Paungmali, A.; Sitilertpisan, P. Hashim, U.F.; Mazlan, M.B.; Nasuha, T.N. Respiratory characteristics of individuals with non-specific low back pain: A cross-sectional study. Nurs Health Sci. 2018, 20, 224-230. 

Point 3: Line 92: Is the scapulocostal syndrome sufficiently defined by just one myofascial trigger point around the scapular region?

Response 3: The criteria was rewriting as shown in the materials and methods and started from page 3 Line 143.

Participants were included if they had experienced pain at the scapular region for longer than 12 weeks (VAS equal to or more than 5 cm) covering at least one MTrPs in the muscles around the scapular region, i.e., the levator scapulae, trapezius, rhomboid, teres, and serratus posterior superior muscles. Trigger points were detected as the presence of tender spot within palpable taut bands of muscle in regions that the patient known as painful. MTrP produce a specific referred pain pattern [22]. Healthy participants were recruited as a control group if they had no history of SCS throughout the 12 months prior.

Point 4: Line 96: Why did the study not exclude those with history of radiotherapy? Patients received radiotherapy can have much reduced chest cage mobility.

Response 4: The exclusion criteria was rewriting followed by the suggestion of reviewer as shown in the materials and methods section and started from page 4 Line 155.

Thank you very much for the valuable suggestion. This study concern history of radiotherapy. Participants in this study did not received radiotherapy. Moreover, authors have been added exclusion criteria, patients received radiotherapy were excluded. This suggestion advantage for this study and the future study.

Point 5: Line 106: Is it calculated by M mode ultrasonography? The right subcostal area is large, and different location of probe placement may greatly impact the diaphragmatic excursion. How did the authors reduce the variations of probe placement among different participants?

Response 5:   The procedures of diaphragmatic mobility was rewriting followed by the suggestion of reviewer as shown in the materials and methods section and started from page 4 Line 165.

     The assessor placed the transducer over the right subcostal area, with the striking angle of the ultrasound towards the cranio-caudal axis to identify the left portal vein branch as a reference point and this is considered to a valid reference point.

Point 6. Line 127: Was your data normally distributed calculated by the Shapiro-Wilk test? If it was not, how could you use independent t test to perform statistical analysis?

Response 6:   The statistical analysis was rewriting followed by the suggestion of reviewer as shown in the materials and methods section and started from page 5 Line 183

All statistical analyses were performed using SPSS version 26 statistical software. Results were presented as mean ± standard deviation (SD). Normality of data was evaluated via the Shapiro-Wilk test. Based on the assumption of normality, independent t-tests were used to compare outcome variables between groups. For all tests, statistical significance threshold was set at p < 0.05. 

Point 7: Line 134: The content in the paragraph should not redundantly duplicate the data in the tables.

Response 7:   The demographic data and baseline clinical characteristics was rewriting followed by the suggestion of reviewer as shown in the results section and started from page 5 Line 190.

3.1.            Demographic data and baseline clinical characteristics

The demographic data and baseline clinical characteristics of SCS and healthy par-ticipants are shown in table 1. Twenty-nine SCS participants (17 female and 12 male) were matched with a twenty-nine-member healthy group (17 female and 12 male). There was no significant difference between demographic data including age, height, weight, and BMI with p > 0.05. in SCS group, the MTrPs was found in levator scapulae, upper tra-pezius, and rhomboid muscles. Moreover, number of participants who exercised in SCS was lower than in the healthy group. 

       Table 1. Demographic data of the sample population at baseline.

Characteristics

SCS group

(n= 29)

Healthy group

(n= 29)

p value

Age (years)

26.86± 4.22

26.86± 4.22

1.000

Gender (female: male)

17: 12

17: 12

Weight (kg)

56.36± 8.57

56.17± 8.48

0.933

Height (cm)

164.00± 9.87

164.55± 9.32

0.828

BMI (kg/m2)

20.86± 1.46

20.64± 1.40

0.566

Affected muscle

     Levator scapualae (%)

10 (34.48%)

0

     Upper trapezius (%)

8 (27.59%)

0

     Rhomboid (%)

11 (37.93)

0

Exercise (yes: no)

13: 16

23: 6

Note: kg: kilogram; cm: centimeter; m: meter

Point 8. Line 141: The authors should use dot plot to present their data grouped by the number of trigger points for each participant. Is there a high correlation between the number of trigger points and the amount of diaphragmatic excursion?

Response 8: Thank you for your suggestion. However, in this study choose the most pain of affected muscle. This study did not show data of number of trigger point. This is limitation of this study. The authors added information about the affected muscle in Table 1. Moreover, we have added more details in limitation of the study as started from page 7 Line 299.

     Moreover, future studies should evaluate the number of trigger points for each participant. There may be a high correlation between the number of trigger points and the amount of diaphragmatic excursion.

Reviewer 2 Report

Comments to the Author 
Healthcare Journal

Manuscript ID: healthcare-1690466

Thank you for the opportunity to review the manuscript entitled: “Selected Respiratory Characteristics in Patients with Scapulocostal Syndrome: A Cross-Sectional Study”.

The authors performed a cross-sectional study to evaluate the respiratory limitation that present patients with scapulocostal syndrome. The study focused on a specific topic in the physical therapy investigation, in order to improve the approach of patient who experience scapulocostal syndrome. In this way, research applications may be of interest for this approach. However, the interest of this study is limited to a reduced number of people, and some revision would be needed. Critical comments for authors are stated below. 

Specific comments

TITLE

- P. 1, line 2. Given the content of the article, you should consider to modify the title. The term “Selected Respiratory Characteristics” may not be enough clear to represent the aim of the study.

ABSTRACT

- P. 1 line 24. Please, in the results section, could you add some numerical/statistical values?

- P. 1 line 27. In the discussion section, you can also focus on the clinical application. 

INTRODUCTION

The introduction tries to states the rationale for the study, but the bibliography provided is limited. In addition, the variables stated represent little explored or unknown areas of the sequels of  those patients.

- P.1, line 33. It could be interesting to include information about the prevalence or  socio-demographic data of this syndrome.

- P.2, line 47. It is not clear the relation between scapular muscles-core muscles and diaphragm, could you go in-depth into the scientific background of it?

- P.2, line 55. It could be interesting to go in-depth into the scientific background of the scapulocostal syndrome and the respiratory repercussion, comparing with similar myofascial pain syndrome, and showing the repercussion in the health status or the quality of life of the patients.

- P.2, line 63. It could be interesting to go in-depth into the scientific background of the  evaluations of lower chest expansion. 

- P.2, line 73. I think the objective should be rewrite in order to do it more adequate to the content of the article.

METHODS

- P.2, line 86. Could you clarify me why the study only included participant aged between 18 and 50 years? It is not justified in the study.

- P.2, line 89, Could you introduce a reference that support the diagnosis criteria of the 

scapulocostal syndrome used by the physiatrist? If these criteria are the same which are referred in the inclusion criteria, please, specify and reference it. To have this syndrome imply to have pain? This is not specify during introduction, nevertheless the methodology specify 12 weeks of pain experience. It is not enough specified for putting it as cut-off, in the case of be a validate cut-off, please you should reference it.

- P.2, line 95. It could be interesting to know if these patients usually practice some exercise or regular physical activity.

- P.3, line 103. Could you clarify me why the study only excluded participant who smoked more than the number of cigarettes referred in the methodology?

- P.3, line 104. I think that to not separate, or at less to not clarify, the myofascial trigger points on muscles scapulae, can have a different repercussion in the thorax and spine mobility. For this reason, It could be interesting to include in the study a table of the exploration of the active myofascial trigger points, in order to facilitate the visualization of them.

- P.3, line 104. If the title specifies “Respiratory Characteristics”, a specific respiratory measure should be included to demonstrate the respiratory affectation of the topic, including a spirometry as gold standard of the respiratory function.  If the focus of the approach of these alterations is not the respiratory repercussions, other variable as pain or quality of life should be included to demonstrate any repercussion of these “respiratory characteristics”, if you want to show the importance of the treatment of it as it is commented in the conclusion.

- P.3, line 105. It would be interesting to clarify the validation of the tools.

RESULTS

- P.4, line 133. Please, could you write the same order of male / female in the text and in the table?

- P.4, Table 1. The values of gender have “:” in the SCS group, nevertheless, Healthy group only have a point.

- P.4, Table 1. Could you include the legend of the table specifying the abbreviations included?

- P.4, Table 2. Could you include the legend of the table specifying the abbreviations included?

DISCUSSION

- P.5, line 162. After the aim of the study, you should compare your results with previous studies. The discussion needs more references.

- P.5, line 181. It could be interesting to discuss why the results of axillar expansion are not significant.

- P.5, line 183. Could you include more bibliography that support the statements provided in the hold discuss section?

- P.5, line 207. Why is assumed that a particular posture can generate an overuse of the muscles? You should clarify it and discuss it in a deep way with the publications of other authors.

Author Response

The school of Physical Therapy,

Faculty of Associated Medical Science,

Khon Kaen University, Khon Kaen 40002,

Thailand Tel / Fax 66-43-202085 Email: [email protected]

Date 6 May 2022

Dear Ms. Alina-Sabina Buglea

Editor-in-Chief of Healthcare– MDPI

Title: Selected respiratory characteristics in patients with scapulocostal syndrome: A cross-sectional study

Authors: Thanaporn Srijessadarak, Preeda Arayawichanon, Jaturat Kanpittaya and Yodchai Boonprakob

I am pleased to submit an original research article entitled " Selected respiratory characteristics in patients with scapulocostal syndrome: A cross-sectional study " for publication in the Healthcare.We investigated diaphragmatic mobility and chest expansion between individuals with and without scapulocostal syndrome. The result showed that diaphragmatic mobility value was lower in the SCS group when compared to healthy participants. Chest expansion at the 4th ICS and xiphoid level in the SCS group was significantly less than in the healthy group. In conclusion, SCS patients presented diaphragmatic mobility and chest expansion less than healthy participants. Therefore, these variables should be investigated in patients with SCS, and SCS treatment should focus on diaphragmatic mobility and chest expansion. We believe these findings will be interest to the reader of your journal.

On behalf of all authors, I would like to sincerely thank you for all kind correspondence and support from the editor. I also thank to valuable time, comments and suggestions from the editor and reviewers helping to improve the quality and clarity of the paper. In this version, the major and minor revisions have been made according to the suggestion as indicated using yellow highlight and explained in the “response to reviewers” file. If there is any further information about my work, please do not hesitate to contact me. I am looking forward to hearing from you very soon.

Yours sincerely,

Yodchai Boonprakob

Response to Reviewer 2 Comments

TITLE

Point 1: P. 1, line 2. Given the content of the article, you should consider to modify the title. The term “Selected Respiratory Characteristics” may not be enough clear to represent the aim of the study.

Response 1: The title was rewriting followed by the suggestion of reviewer as shown in the page 1 Line 2.

Diaphragmatic Mobility and Chest Expansion in Patients with Scapulocostal Syndrome: A Cross-Sectional Study

ABSTRACT

Point 2: P. 1 line 24. Please, in the results section, could you add some numerical/statistical values?

Response 2: The introduction was rewriting followed by the suggestion of reviewer as shown in the introduction section and started from page 1 Line 23

The DM value in the SCS group was 46.24± 7.26 mm, whereas in the healthy group it was 54.18± 9.74 mm. DM value was lower in the SCS group when compared to healthy participants (p < 0.05). Chest expansion at the axilla, the fourth intercostal space (4th ICS), and xiphoid level in the SCS group was 7.26± 1.13, 6.83± 0.94, and 6.86±1.25, respectively, while chest expansion at the axilla, 4th ICS, and xiphoid level in the healthy group was 7.92±1.39, 7.54±1.43, and 8.13±1.32, respectively. Chest expansion at the 4th ICS, and xiphoid level in the SCS group was significantly lower than in the healthy group (p < 0.05).

Point 3: P. 1 line 27. In the discussion section, you can also focus on the clinical application.

Response 3: The clinical application was rewriting followed by the suggestion of reviewer as shown in the introduction section and started from page 1 Line 29

Patients with SCS presented a decrease in diaphragmatic mobility and chest expansion. Therefore, SCS treatment program ought to add breathing exercise for improving lung expansion.

INTRODUCTION

Point 4: P.1, line 33. It could be interesting to include information about the prevalence or socio-demographic data of this syndrome.

Response 4: We have added more details in the introduction section as shown in page 1 Line 38.

Scapulocostal syndrome (SCS) is a chronic myofascial pain syndrome affecting the thoracic and scapular region. SCS is a chronic condition which is pain that is ongoing and usually lasts longer than three months [1]. The thoracic spine pain prevalence data of 1 year ranged from 3.0-55.0% [2]. The lifetime upper back prevalence was 59.5% [3]. Moreover, the incidence of SCS was found in middle-aged between 18 to 60 years, and especially in adult working population. This syndrome was found in females more than males.

Point 5: P.2, line 47. It is not clear the relation between scapular muscles-core muscles and diaphragm, could you go in-depth into the scientific background of it?

Response 5: We have added more details in the introduction section as shown in page 2 Line 56.

The core comprises various different muscles which stabilize the shoulders, the pelvis and the spine and provides a base for movement in the limbs. Major core muscles include transversus abdominus in the anterior, multifidus in the posterior, pelvic floor in the inferior and diaphragm at the superior. Minor core muscles are the latissimus dorsi, gluteus maximus and the trapezius. All of these muscles connect directly or indirectly to the thoracolumbar fascia and spinal column, which attach the upper and lower extrem-ities. The core is appreciated as the center of the functional kinetic chain. The core muscles are initiated through a feed-forward mechanism shortly before movements of the upper and lower limb to performance as a base which skilled movements can be performed. This feed-forward mechanism is essential for attaining mobility and stability of the extremities. These findings encouragement the theory that movement control and stability are de-veloped in a core-to-extremity (proximal-distal) and cephalo-caudal manner (head-to-toe) [7].

Point 6: P.2, line 55. It could be interesting to go in-depth into the scientific background of the scapulocostal syndrome and the respiratory repercussion, comparing with similar myofascial pain syndrome, and showing the repercussion in the health status or the quality of life of the patients.

Response 6: We have added more details in the introduction section as shown in page 2 Line 70.

Recently, respiration or function of the diaphragm muscle was evaluated in many myofascial pain syndromes including neck pain, temporomandibular joint pain, low back pain, and lumbopelvic pain [8]. Janssens and coworkers found that participants with low back pain presented more diaphragm fatigability as compared to healthy participants [9]. Mohan et al. presented that diaphragmatic movement and respiratory muscle endurance were poorer in the nonspecific lower back pain group than in the healthy group [10]. Recently, Calvo-Lobo and colleague reported that participants with lumbopelvic pain presented a reduced diaphragm thickness compared to healthy matched-paired partic-ipants [11]. The diaphragm muscle is one of core stabilizer which related with postural control. Ineffective diaphragm muscle led to poor postural control, poor balance, adjusted proprioception, and ineffective motor control. Additionally, this led to abnormal breathing which is fabricated by the accessory muscles of respiration involving ster-nocleidomastoid, upper trapezius, and scalene muscles. Over‐action of these accessory muscles caused neck pain, scapular dyskinesis, and trigger point formation [8]. Moreover, Previous study found that FHP was correlated with a decrease in respiratory muscle strength in patient with chronic neck pain. This caused by morphological and biome-chanical changes in thoracic cage. To clarify, FHP led to expansion of upper chest and narrowing of lower chest, which limits lower chest expansion. Moreover, FHP contributed to abdominal muscle shortening, resulting in a decrease of the anteroposterior diameter of the lower chest. This caused of limit diaphragmatic mobility [12].

Point 7: P.2, line 63. It could be interesting to go in-depth into the scientific background of the  evaluations of lower chest expansion.  

Response 7: We have added more details in the introduction section as shown in page 3 Line 99.

Chest expansion used to assess rib cage mobility and was established to be associated with lung volume. An association between upper or lower CE and maximal inspiratory pressure was earlier determined in patients with fibromyalgia and osteoporosis. Ana-tomical reference for upper CE includes the fourth intercostal space, axillary level, and 5th thoracic vertebrae, and lower CE include xiphoid level and 10th thoracic vertebrae. As it is measured using a cloth tape measure, it is a simple, inexpensive, and noninvasive equipment for evaluating chest mobility [16]. Previous study found that lower chest expansion correlated with diaphragmatic mobility (r=0.74, p-value= 0.001) [17]. Conse-quently, real time ultrasound and cloth tape measure are a suitable instrument to evaluate the mobility of the diaphragm and chest expansion in this study.

Point 8: P.2, line 73. I think the objective should be rewrite in order to do it more adequate to the content of the article.

Response 8: The objective was rewriting as shown in the abstarct and the introduction section and started from page 3 Line 119.

The purpose of this study was to investigate characteristic of diaphragmatic mobility and chest expansion in patients with SCS.

METHODS

Point 9: P.2, line 86. Could you clarify me why the study only included participant aged between 18 and 50 years? It is not justified in the study.

Response 9: The inclusion criteria of this study were based on previous study. This syndrome can be present in student and worker who have poor posture aged between 18 and 60 years (Ormandy 1994). However, people older than 50 years is age-related macular degeneration (AMD). Therefore, this study included participant aged between 18 and 50 years. The patient age over 50 years old should be specifically stuied in future study because some patient may be found underlying disease.

Point 10: P.2, line 89, Could you introduce a reference that support the diagnosis criteria of the scapulocostal syndrome used by the physiatrist? If these criteria are the same which are referred in the inclusion criteria, please, specify and reference it. To have this syndrome imply to have pain? This is not specify during introduction, nevertheless the methodology specify 12 weeks of pain experience. It is not enough specified for putting it as cut-off, in the case of be a validate cut-off, please you should reference it.

Response 10: We have added more details in the methods section as shown in page 3 Line 142.

Participants were included if they had experienced pain at the scapular region for longer than 12 weeks (VAS equal to or more than 5 cm) covering at least one MTrPs in the muscles around the scapular region, i.e., the levator scapulae, trapezius, rhomboid, teres, and serratus posterior superior muscles. Trigger points were detected as the presence of tender spot within palpable taut bands of muscle in regions that the patient known as painful. MTrP produce a specific referred pain pattern [22]. Healthy participants were recruited as a control group if they had no history of SCS throughout the 12 months prior.

Point 11: P.2, line 95. It could be interesting to know if these patients usually practice some exercise or regular physical activity.

Response 11: We have added more details in the table 1 as shown in Table 1

Table 1. Demographic data of the sample population at baseline.

Characteristics

SCS group

(n= 29)

Healthy group

(n= 29)

p value

Age (years)

26.86± 4.22

26.86± 4.22

1.000

Gender (female: male)

17: 12

17: 12

Weight (kg)

56.36± 8.57

56.17± 8.48

0.933

Height (cm)

164.00± 9.87

164.55± 9.32

0.828

BMI (kg/m2)

20.86± 1.46

20.64± 1.40

0.566

Affected muscle

     Levator scapualae (%)

10 (34.48%)

0

     Upper trapezius (%)

8 (27.59%)

0

     Rhomboid (%)

11 (37.93)

0

Exercise (yes: no)

13: 16

23: 6

Note: kg: kilogram; cm: centimeter; m: meter

Point 12: P.3, line 103. Could you clarify me why the study only excluded participant who smoked more than the number of cigarettes referred in the methodology?

Response 12: Participant who smoked weas exclude. This information was rewriting to clearly explanation as shown in page 4 Line 157.

The exclusion criteria incorporated any of the following disorders: history of de-generative shoulder joint disease, rotator cuff disease, adhesive shoulder capsulitis, cer-vical radiculopathy with facet joint dysfunction and/or intervertebral disc herniation, lumbar intervertebral disc herniation, lumbar stenosis, lumbar spondylosis, lumbar spondylolisthesis, history of radiotherapy, chronic respiratory diseases (chronic obstruc-tive pulmonary disease (COPD), asthma, occupational lung diseases or pulmonary hy-pertension), Smoker, and ex-smoker.

Point 13: P.3, line 104. I think that to not separate, or at less to not clarify, the myofascial trigger points on muscles scapulae, can have a different repercussion in the thorax and spine mobility. For this reason, It could be interesting to include in the study a table of the exploration of the active myofascial trigger points, in order to facilitate the visualization of them.

Response 13: We have added more details in the table 1 as follows:

Table 1. Demographic data of the sample population at baseline.

Characteristics

SCS group

(n= 29)

Healthy group

(n= 29)

p value

Age (years)

26.86± 4.22

26.86± 4.22

1.000

Gender (female: male)

17: 12

17: 12

Weight (kg)

56.36± 8.57

56.17± 8.48

0.933

Height (cm)

164.00± 9.87

164.55± 9.32

0.828

BMI (kg/m2)

20.86± 1.46

20.64± 1.40

0.566

Affected muscle

     Levator scapualae (%)

10 (34.48%)

0

     Upper trapezius (%)

8 (27.59%)

0

     Rhomboid (%)

11 (37.93)

0

Exercise (yes: no)

13: 16

23: 6

Note: kg: kilogram; cm: centimeter; m: meter

Point 14: P.3, line 104. If the title specifies “Respiratory Characteristics”, a specific respiratory measure should be included to demonstrate the respiratory affectation of the topic, including a spirometry as gold standard of the respiratory function.  If the focus of the approach of these alterations is not the respiratory repercussions, other variable as pain or quality of life should be included to demonstrate any repercussion of these “respiratory characteristics”, if you want to show the importance of the treatment of it as it is commented in the conclusion.

Response 14: This study did not measure respiratory characteristic by spirometer. This is limitation of this study. We have added more details in the limitation and started from page 7 Line 298.

Point 15: P.3, line 105. It would be interesting to clarify the validation of the tools.

Response 15: We have added more details in the materials and methods and started from page 4 Line 161.

This tool is valuable to accurately assess diaphragm mobility. RTUS showed high current validity (r=0.78 to r=0.83) [23].

RESULTS

Point 16: P.4, line 133. Please, could you write the same order of male / female in the text and in the table?

Response 16: The data was checked. The order of male / female was rewriting followed by the suggestion of reviewer as shown in result section frome page 5 Line 191 and the Table 1.

Point 17: P.4, Table 1. The values of gender have “:” in the SCS group, nevertheless, Healthy group only have a point.

Response 17: The data was checked. The symbol (.) was deleted and replaced by the symbol (:) as shown in Table 1.

Point 18: P.4, Table 1. Could you include the legend of the table specifying the abbreviations included?

Response 18: The abbreviations was rewriting followed by the suggestion of reviewer as shown in the Table 1.

From point 16-18, we have added more details in the table 1 as follows:

The demographic data and baseline clinical characteristics of SCS and healthy participants are shown in table 1. Twenty-nine SCS participants (17 female and 12 male) were matched with a twenty-nine-member healthy group (17 female and 12 male). There was no significant difference between demographic data including age, height, weight, and BMI with p > 0.05. in SCS group, the MTrPs was found in levator scapulae, upper trapezius, and rhomboid muscles. Moreover, number of participants who exercised in SCS was lower than in the healthy group. 

Table 1. Demographic data of the sample population at baseline.

Characteristics

SCS group

(n= 29)

Healthy group

(n= 29)

p value

Age (years)

26.86± 4.22

26.86± 4.22

1.000

Gender (female: male)

17: 12

17: 12

Weight (kg)

56.36± 8.57

56.17± 8.48

0.933

Height (cm)

164.00± 9.87

164.55± 9.32

0.828

BMI (kg/m2)

20.86± 1.46

20.64± 1.40

0.566

Affected muscle

     Levator scapualae (%)

10 (34.48%)

0

     Upper trapezius (%)

8 (27.59%)

0

     Rhomboid (%)

11 (37.93)

0

Exercise (yes: no)

13: 16

23: 6

Note: kg: kilogram; cm: centimeter; m: meter

Point 19: P.4, Table 2. Could you include the legend of the table specifying the abbreviations included?

Response 19: The abbreviations was rewriting followed by the suggestion of reviewer as shown in the Table 2.

Table 2. Diaphragmatic mobility and chest expansion between SCS group and healthy group.

Characteristics

SCS group

(n= 29)

Healthy group (n= 29)

Difference

(95% CI)

p value

Diaphragmatic

mobility (mm)

46.24± 7.26

54.18± 9.74

-7.94 (-12.46 to

-3.41)

0.001*

Chest expansion (cm)

1. Axilla: mean

7.26± 1.13

7.92± 1.39

-0.66 (-1.33 to 0.01)

0.053

2. 4th ICS: mean

6.83± 0.94

7.54± 1.43

-0.71 (-1.34 to

-0.07)

0.031*

3. Xiphoid: mean

6.86± 1.25

8.13± 1.32

-1.27 (-1.95 to

-0.60)

<0.001*

* Statistically significant (p < 0.05)

Note: mm: millimeter; cm: centimeter

DISCUSSION

Point 20: P.5, line 162. After the aim of the study, you should compare your results with previous studies. The discussion needs more references.

Response 20: We have added more details in the discussion as shown in page 6 Line 216.

Reduced diaphragmatic mobility and chest expansion in SCS patient was found in this study. To our knowledge, this is the first study to evaluate diaphragmatic mobility and chest expansion in patients with SCS. However, evaluation of respiratory characteris had been evaluated in patient with neck pain, low back pain, and lumbopelvic pain. The possible mechanism for explaining these results will be discuss in a logical way as follow:

Point 21: P.5, line 181. It could be interesting to discuss why the results of axillar expansion are not significant.

Response 21: We have added more details in the discussion as shown in page 6 Line 223.

Based on anatomical, diaphragm muscle located in lower chest. During inhalation, the abdomen moves forward as the lower six ribs laterally expand, elevate, and rotate upward, comparative with the spine. The sternum and remainder of the thoracic cavity move anteriorly and superiorly, expanding chest volume as the diaphragm descends and produces a negative pressure gradient to draw air into the lungs. During expiration, the diaphragm relaxes and returns to a dome shape. Therefore, chest expansion in this study limited expansion of the middle and lower chest while the upper chest is no different.

Point 22: P.5, line 183. Could you include more bibliography that support the statements provided in the hold discuss section?

Response 22: We have added more details in the discussion section as shown in Line and reference as shown in page 6 Line 245.

Point 23: P.5, line 207. Why is assumed that a particular posture can generate an overuse of the muscles? You should clarify it and discuss it in a deep way with the publications of other authors.

Response 23: We have added more details in the discussion as shown in page 7 Line 269.

The characteristics are a prolonged sitting posture, an awkward posture, and a repetitive movement. An awkward posture has defined the combination between forward head posture and round shoulder simultaneously. The cervical and thoracic spine are held in flexion that leads to stimulating over contraction of the dorsal muscles, such as neck ex-tensor and upper back extensor muscles. Thus, an awkward posture may contribute to neck and shoulder pain. These factors may be caused by muscle imbalance of the upper back or upper crossed syndrome. It shows the tightness of pectoral and neck extensor muscles, whereas weakness of neck flexor and interscapular muscles may occur simul-taneously. The weakness is usually caused by guarding, without atrophy or neuro-physiologic evidence of denervation on electromyography (Abrams, 2011). The weakness usually involves scapular stabilizers that consist of trapezius and serratus anterior mus-cles. In addition, weakness of the levator scapulae or rhomboids muscle is presented with MTrPs [32].

Reviewer 3 Report

I think that the article is well written, with all sections well described.  I think that the article is publishable in Healthcare journal but I have some minor comments for the authors:

  • It is not clear if you are including smokers or not in the participants section. If smokers are not included it is enough to mention that one exclusion criteria is smoking.
  • In the discussion section is mentioned that there were no differences for chest expansion at the axilla level. Could you discuss if this was an expected result? Same results have been described before?

Author Response

The school of Physical Therapy,

Faculty of Associated Medical Science,

Khon Kaen University, Khon Kaen 40002,

Thailand Tel / Fax 66-43-202085 Email: [email protected]

Date 6 May 2022

Dear Ms. Alina-Sabina Buglea

Editor-in-Chief of Healthcare– MDPI

Title: Selected respiratory characteristics in patients with scapulocostal syndrome: A cross-sectional study

Authors: Thanaporn Srijessadarak, Preeda Arayawichanon, Jaturat Kanpittaya and Yodchai Boonprakob

I am pleased to submit an original research article entitled " Selected respiratory characteristics in patients with scapulocostal syndrome: A cross-sectional study " for publication in the Healthcare.We investigated diaphragmatic mobility and chest expansion between individuals with and without scapulocostal syndrome. The result showed that diaphragmatic mobility value was lower in the SCS group when compared to healthy participants. Chest expansion at the 4th ICS and xiphoid level in the SCS group was significantly less than in the healthy group. In conclusion, SCS patients presented diaphragmatic mobility and chest expansion less than healthy participants. Therefore, these variables should be investigated in patients with SCS, and SCS treatment should focus on diaphragmatic mobility and chest expansion. We believe these findings will be interest to the reader of your journal.

On behalf of all authors, I would like to sincerely thank you for all kind correspondence and support from the editor. I also thank to valuable time, comments and suggestions from the editor and reviewers helping to improve the quality and clarity of the paper. In this version, the major and minor revisions have been made according to the suggestion as indicated using yellow highlight and explained in the “response to reviewers” file. If there is any further information about my work, please do not hesitate to contact me. I am looking forward to hearing from you very soon.

Yours sincerely,

Yodchai Boonprakob

Response to Reviewer 3 Comments

Point 1: It is not clear if you are including smokers or not in the participants section. If smokers are not included it is enough to mention that one exclusion criteria is smoking.

Response 1: This section was rewriting followed by the suggestion of the reviewer as shown in exclusion criteria in page 4 Line 157.

The exclusion criteria incorporated any of the following disorders: history of de-generative shoulder joint disease, rotator cuff disease, adhesive shoulder capsulitis, cervical radiculopathy with facet joint dysfunction and/or intervertebral disc herniation, lumbar intervertebral disc herniation, lumbar stenosis, lumbar spondylosis, lumbar spondylolisthesis, history of radiotherapy, chronic respiratory diseases (chronic obstructive pulmonary disease (COPD), asthma, occupational lung diseases or pulmonary hypertension), Smoker, and ex-smoker.

Point 2: In the discussion section is mentioned that there were no differences for chest expansion at the axilla level. Could you discuss if this was an expected result? Same results have been described before?

Response 2: This information was more explain in discussion to clearly explanation as shown in page 6 Line 223.

Based on anatomical, diaphragm muscle located in lower chest. During inhalation, the abdomen moves forward as the lower six ribs laterally expand, elevate, and rotate up-ward, comparative with the spine. The sternum and remainder of the thoracic cavity move anteriorly and superiorly, expanding chest volume as the diaphragm descends and produces a negative pressure gradient to draw air into the lungs. During expiration, the diaphragm relaxes and returns to a dome shape [8]. Therefore, chest expansion in this study limited expansion of the middle and lower chest while the upper chest is no dif-ferent.

Reviewer 4 Report

All information and suggestion I added in addition file.

Best regards

Author Response

The school of Physical Therapy,

Faculty of Associated Medical Science,

Khon Kaen University, Khon Kaen 40002,

Thailand Tel / Fax 66-43-202085 Email: [email protected]

Date 6 May 2022

Dear Ms. Alina-Sabina Buglea

Editor-in-Chief of Healthcare– MDPI

Title: Selected respiratory characteristics in patients with scapulocostal syndrome: A cross-sectional study

Authors: Thanaporn Srijessadarak, Preeda Arayawichanon, Jaturat Kanpittaya and Yodchai Boonprakob

I am pleased to submit an original research article entitled " Selected respiratory characteristics in patients with scapulocostal syndrome: A cross-sectional study " for publication in the Healthcare.We investigated diaphragmatic mobility and chest expansion between individuals with and without scapulocostal syndrome. The result showed that diaphragmatic mobility value was lower in the SCS group when compared to healthy participants. Chest expansion at the 4th ICS and xiphoid level in the SCS group was significantly less than in the healthy group. In conclusion, SCS patients presented diaphragmatic mobility and chest expansion less than healthy participants. Therefore, these variables should be investigated in patients with SCS, and SCS treatment should focus on diaphragmatic mobility and chest expansion. We believe these findings will be interest to the reader of your journal.

On behalf of all authors, I would like to sincerely thank you for all kind correspondence and support from the editor. I also thank to valuable time, comments and suggestions from the editor and reviewers helping to improve the quality and clarity of the paper. In this version, the major and minor revisions have been made according to the suggestion as indicated using yellow highlight and explained in the “response to reviewers” file. If there is any further information about my work, please do not hesitate to contact me. I am looking forward to hearing from you very soon.

Yours sincerely,

Yodchai Boonprakob

Response to Reviewer 4 Comments

Point 1: 1. 24. The '4th ICS' abbreviation is used which has not been explained previously - neither from the scientific nor the clinical point of view. I predict that some readers might find it unclear so I would use the full terminology of “4th Inter Costal”.

Response 1: The introduction was rewriting followed by the suggestion of reviewer as shown in the Introduction section and started from page 1 Line 26.

The DM value in the SCS group was 46.24± 7.26 mm, whereas in the healthy group it was 54.18± 9.74 mm. DM value was lower in the SCS group when compared to healthy participants (p < 0.05). Chest expansion at the axilla, the fourth intercostal space (4th ICS), and xiphoid level in the SCS group was 7.26± 1.13, 6.83± 0.94, and 6.86±1.25, respectively, while chest expansion at the axilla, 4th ICS, and xiphoid level in the healthy group was 7.92±1.39, 7.54±1.43, and 8.13±1.32, respec-tively.

Point 2: 131. I see the information enclosed in the chapter 3.1 (inclusive of tab.1) as facts of the researched people so I would therefore move this to the chapter 2.2.

Response 2: The information enclosed in the chapter 3.1 was rewriting followed by the suggestion of reviewer as shown in the result section and started from page 5 Line 190.

The demographic data and baseline clinical characteristics of SCS and healthy par-ticipants are shown in table 1. Twenty-nine SCS participants (17 female and 12 male) were matched with a twenty-nine-member healthy group (17 female and 12 male). There was no significant difference between demographic data including age, height, weight, and BMI with p > 0.05. in SCS group, the MTrPs was found in levator scapulae, upper tra-pezius, and rhomboid muscles. Moreover, number of participants who exercised in SCS was lower than in the healthy group. 

Point 3: Tab 1. presents a range of punctuation by the gender specification. On one occasion a ':' is used, on another- a '.' is used. I would suggest the use of same in both.

Response 3: The data was checked. The symbol (.) was deleted and replaced by the symbol (:) as shown in Table 1

Table 1. Demographic data of the sample population at baseline.

Characteristics

SCS group

(n= 29)

Healthy group

(n= 29)

p value

Age (years)

26.86± 4.22

26.86± 4.22

1.000

Gender (female: male)

17: 12

17: 12

Weight (kg)

56.36± 8.57

56.17± 8.48

0.933

Height (cm)

164.00± 9.87

164.55± 9.32

0.828

BMI (kg/m2)

20.86± 1.46

20.64± 1.40

0.566

Affected muscle

     Levator scapualae

10

     Upper trapezius

8

     Rhomboid

11

Exercise (yes: no)

13: 16

23: 6

Note: kg: kilogram; cm: centimeter; m: meter

Point 4: 147. A strange sign appears instead of '±'. I think chart 2, showcasing same data as presented in table 2, serves no point.

Response 4: The data was checked. The symbol was replaced by the symbol (±) as shown in result section and started from page 5 Line 204.  

Chest expansion at the axilla, 4th ICS, and xiphoid level in the SCS group was 7.26± 1.13, 6.83± 0.94, and 6.86±1.25, respectively, while chest expansion at the axilla, 4th ICS, and xiphoid level in the healthy group was 7.92±1.39, 7.54±1.43, and 8.13±1.32, respectively.

Round 2

Reviewer 1 Report

  1. Page 3 Line 128: Regarding to sample size calculation, the authors failed to mention the software used to calculate the required sample size. Furthermore, according to the data provided by the authors, the diaphragm mobility in participants with nonspecific low back pain was 45.09±89 mm, and that of healthy controls was 50.09±9.18 mm, the effect size should be 0.524, and a total of 118 subjects should be included in this study. Please refer to the reference figure of sample size calculation. I’m afraid that the included subjects in this study were far from sufficient.
  2. Page 4 Line 149: Was the typical referred pain pattern a necessary criterion to make the diagnosis of MTrPs? As I know, not all patients with a MTrPs have exactly the same referred pain pattern.
  3. Page 4 Line 165: M mode ultrasonography should be mentioned in the paragraph. The authors identified the left portal vein branch as a reference point and considered it to a valid reference point. Do the authors have any reference to support this statement? In Figure 1, the left portal vein branch was not clearly visualized in the healthy subject. Was it a mistaken example to place the ultrasound transducer?
  4. Page 5 Line 184: The authors should not perform statistical analysis “based on the assumption of normality”. The degree of distributed normality should be formally examined by the Shapiro-Wilk test.

Reviewer 2 Report

INTRODUCTION

More references need to be included.

- P.2, line 83. When you said “previous study”, which study are you referred? Please, can you introduce a reference?

METHODS

- P.3, line 134. The sentence “Previous study report…” has other source.

RESULTS

- P.4, Table 1. The legend of the table should include: “Data are expressed as mean ± standard deviation”

- P.4, Table 1. The legend of the table should include the meaning of “SCS”.

- P.4, Table 2. The legend of the table should include: “Data are expressed as mean ± standard deviation”

- P.4, Table 2. The legend of the table should include the meaning of “ICS”.

DISCUSSION

- P.7, line 278. The reference “Abrams,2011” is in a wrong format.

- P.5, line 183. Could you include more bibliography that support the statements provided in the hold discuss section?
